# PSEN1 E280A Cholinergic-like Neurons and Cerebral Spheroids Derived from Mesenchymal Stromal Cells and from Induced Pluripotent Stem Cells Are Neuropathologically Equivalent

**DOI:** 10.3390/ijms24108957

**Published:** 2023-05-18

**Authors:** Miguel Mendivil-Perez, Carlos Velez-Pardo, Francisco Lopera, Kenneth S. Kosik, Marlene Jimenez-Del-Rio

**Affiliations:** 1Neuroscience Research Group, Medical Research Institute, Faculty of Medicine, University of Antioquia (UdeA), Calle 70 No. 52-21, Calle 62#52-59, Building 1, Room 412, SIU, Medellin 050010, Colombia; miguel.mendivil@udea.edu.co (M.M.-P.); calberto.velez@udea.edu.co (C.V.-P.); francisco.lopera@gna.org.co (F.L.); 2Neuroscience Research Institute, Department of Molecular Cellular Developmental Biology, University of California, Santa Barbara, CA 93106, USA; kenneth.kosik@lifesci.ucsb.edu

**Keywords:** Alzheimer, apoptosis, E280a, iPSCs, presenilin, mutant, mesenchymal stromal

## Abstract

Alzheimer’s disease (AD) is a chronic neurological condition characterized by the severe loss of cholinergic neurons. Currently, the incomplete understanding of the loss of neurons has prevented curative treatments for familial AD (FAD). Therefore, modeling FAD in vitro is essential for studying cholinergic vulnerability. Moreover, to expedite the discovery of disease-modifying therapies that delay the onset and slow the progression of AD, we depend on trustworthy disease models. Although highly informative, induced pluripotent stem cell (iPSCs)-derived cholinergic neurons (ChNs) are time-consuming, not cost-effective, and labor-intensive. Other sources for AD modeling are urgently needed. Wild-type and presenilin (PSEN)1 p.E280A fibroblast-derived iPSCs, menstrual blood-derived menstrual stromal cells (MenSCs), and umbilical cord-derived Wharton Jelly’s mesenchymal stromal cells (WJ-MSCs) were cultured in *Cholinergic-N-Run* and *Fast-N-Spheres V2* medium to obtain WT and PSEN 1 E280A cholinergic-like neurons (ChLNs, 2D) and cerebroid spheroids (CSs, 3D), respectively, and to evaluate whether ChLNs/CSs can reproduce FAD pathology. We found that irrespective of tissue source, ChLNs/CSs successfully recapitulated the AD phenotype. PSEN 1 E280A ChLNs/CSs show accumulation of iAPPβ fragments, produce eAβ42, present TAU phosphorylation, display OS markers (e.g., oxDJ-1, p-JUN), show loss of ΔΨ_m_, exhibit cell death markers (e.g., TP53, PUMA, CASP3), and demonstrate dysfunctional Ca^2+^ influx response to ACh stimuli. However, PSEN 1 E280A 2D and 3D cells derived from MenSCs and WJ-MSCs can reproduce FAD neuropathology more efficiently and faster (11 days) than ChLNs derived from mutant iPSCs (35 days). Mechanistically, MenSCs and WJ-MSCs are equivalent cell types to iPSCs for reproducing FAD in vitro.

## 1. Introduction

Alzheimer’s disease (AD) is a progressive and chronic neurological condition characterized by loss of memory due to vulnerability and the severe loss of cholinergic neurons from the nucleus basalis magnocellular of Meynert and cholinergic projections to the cortex and hippocampus [1,2]. AD is biologically described not only by the intracellular accumulation of amyloid-beta (iAβ), extracellular amyloid-β (eAβ)-containing plaques, and intracellular hyperphosphorylated tau-containing neurofibrillary tangles [3], but also by dysfunction in the highly interrelated endosomal and lysosomal clearance pathways and loss of synaptic homeostasis [4]. However, it is not yet clear how iAβ and/or eAβ might signal cell death in AD [5]. Therefore, the incomplete understanding of the loss of cholinergic neurons has prevented curative treatments for AD [6]. This situation is even further aggravated by the appearance of early-onset familial AD (FAD) caused by mutations in one or more of at least three genes known as amyloid-beta precursor protein (APP), presenilin 1 (PSEN 1), and presenilin 2 (PSEN2) (https://www.alzforum.org/mutations, accessed on 7 May 2023). Specifically, most mutations in the PSEN1 gene, which codes for the catalytic component of γ-secretase responsible for APP processing [7,8], result in the overproduction of Aβ e.g., the 42-amino acid Aβ isoform [9]. Interestingly, the Glu280Ala (p. E280A, c.839A > C, exon 8) mutation in PSEN 1 is by far the most common and well-characterized causative mutation of FAD, affecting a large population localized in Antioquia, Colombia ([10]; https://www.alzforum.org/mutations/psen1-e280a-paisa, accessed on 7 May 2023). Unfortunately, despite several efforts [11,12], there are no efficient therapies against PSEN 1 E280A FAD. These insidious outcomes have encouraged us, and others, to model the PSEN 1 FAD in vitro.

Modeling FAD with the induction of somatic cells (e.g., fibroblasts) into stem cells (e.g., human induced pluripotent stem cells, hiPSCs) has provided an important biological tool to potentially recreate the neuropathology of the disorder [13]. Indeed, several groups have generated PSEN 1 mutant iPSCs-derived neuronal cells that exhibit elevated Aβ42 generation and increased TAU phosphorylation (e.g., [14,15,16]). Remarkably, an iPSC line derived from a patient with early-onset disease carrying the E280A mutation in the PSEN 1 gene was created [17]. Furthermore, the mutation was introduced into an iPSC line from a healthy individual using the CRISPR-Cas9 technology [18]. Although PSEN 1 E280A iPSCs-derived neurons produce increased levels of eAβ42 [19], no data are available to determine whether mutant iPSCs-derived neurons also display other typical neuropathological hallmarks of FAD, such as accumulation of iAβ, oxidative stress (OS), and neuronal cell death markers, as well as TAU phosphorylation (p-TAU).

Obtaining iPSCs from patients bearing PSEN1 mutations is appealing; however, the isolation and purification procedures are technically challenging, expensive, time-consuming, and labor-intensive. Furthermore, the protocol for the differentiation of iPSCs into neuronal progenitor cells (NPCs), which can subsequently be patterned to different neuronal lineages, such as cholinergic, is time consuming, taking at least 35 days [20]. Alternatively, human mesenchymal stromal (stem) cells (MSCs) derived from menstrual blood (MenB) and Wharton’s jelly tissue (WJ) might be easier to obtain, less expensive, and ready-to-use [21,22,23,24]. The MSCs are multipotent cells that can transdifferentiate into ectodermal lineage cells, such as oligodendrocyte progenitor-like cells [25], nerve-like cells [26], and cholinergic-like neurons [27,28], among others. Indeed, by using the *Cholinergic-N-Run* and *Fast-N-Spheres V2* mediums, we have recreated the molecular pathogenesis of PSEN1 E280A mutation FAD in planar (2D) ChLNs derived from umbilical-cord WJ-MSCs [29] and in 3D (cerebral spheroids) derived from menstrual mesenchymal stromal cells (MenSCs) [30]. The PSEN1 E280A ChLNs and CSs exhibit: (i) early intracellular accumulation of AβPP fragments (iAβPPf, but not Aβ42 peptide); (ii) early oxidized DJ-1 at residue Cys^106^SO_3_, indicative of OS; (iii) elevated p-TAU; (iv) loss of the mitochondrial membrane potential (ΔΨ_m_), (v) displayed cell death markers such as activation of caspase 3 (CASP3), and DNA fragmentation; (vi) generated high amounts of eAβ42; and (vii) showed Ca^2+^ influx alteration in response to acetylcholine (ACh) stimuli compared to wild-type ChLNs and CSs [29,30]. These findings in ChLNs and CSs show great promise for modeling human FAD in vitro for regenerative medicine and identifying therapeutic treatments for AD.

The aim of the present study was to determine whether 2D and 3D MenSCs- and umbilical-cord Wharton Jelly’s MSCs-derived PSEN1 E280A ChLNs and CSs were neuropathologically equivalent to PSEN1 E280A ChLNs and CSs resulting from iPSC-derived neural progenitor cells. Therefore, accumulation of iAβ42, production of eAβ42, phosphorylation TAU, presence of OS makers (e.g., oxDJ-1, p-JUN), loss of ΔΨ_m_, cell death markers (e.g., TP53, PUMA, CASP3), and Ca^2+^ influx response to ACh were assessed in PSEN1 E280A ChLNs and CSs derived from iPSCs, MenSCs, and WJ-MSCs. We demonstrate for the first time that FAD PSEN1 E280A pathology can be recapitulated in WJ-MSC- and MenSCs-derived ChLNs and CSs in about 11 days, similar to PSEN1 E280A ChLNs derived from iPSCs-NPC in 35 days.

## 2. Results

### 2.1. Wild-Type and PSEN 1 E280A MenSCs, and WJ-MSCs Express Comparable Cellular Pluripotential Markers as iPSCs

We wanted to first confirm that wild-type (WT) and mutant iPSCs, MenSCs, and WJ-MSCs displayed typical cellular pluripotential markers. As shown in Figure 1, WT (Figure 1A) and PSEN 1 E280A iPSCs (Figure 1B), MenSCs (Figure 1C,D), and WJ-MSCs (Figure 1E,F) expressed not only the pluripotent transcription factor octamer binding transcription factor 4 (OCT4, Figure 1A–F), but also the transcription factor Sex determining Region Y-box 2 (SOX 2, Figure 1G–L), Homeobox transcription factor Nanog (NANOG, Figure 1M–R), and Krüppel-like factor 2 (KLF, Figure 1S–X), which are essential for pluripotency and self-renewal of stem cells (Figure 1Y,Z,AA,AB). As expected, WT and PSEN 1 E280A iPSCs, MenSCs, and WJ-MSCs differentiated into mesoderm (Figure 1AC–AH), and ectoderm (Figure 1AI–AN) germinal layers during embryonic development according to the presence of vimentin and Nestin markers, respectively; however, the endoderm layer was only present in iPSCs (Figure 1AO–AP) but absent in both WT and mutant MenSCs and WJ-MSCs (Figure 1AQ–AT) according to the endodermal marker C-X-C chemokine receptor type 4 (CXCR4), which is a specific marker for stromal-derived-factor-1.

### 2.2. Wild-Type and PSEN 1 E280A MenSCs and WJ-MSCs Express Neuronal Stem Markers as iPSC-Derived NPC

We initially generated WT and PSEN 1 E280A iPSC-derived neural progenitor cells (NPC) using a well-established stepwise protocol lasting 27 days (Figure 2A,B). The iPSCs (Figure 2C,D) were successively exposed to different culture formulae (Section 4) to obtain embryonic bodies (EB, Figure 2E,F), pre-NPC (Figure 2G,H), and NPC (Figure 2I,J). No evident morphological differences were observed between WT and mutant cells at the different stages of cellular differentiation process (Figure 2C,E,G,I versus Figure 2D,F,H,J).

Then, we evaluated whether WT and PSEN 1 E280A iPSC-derived NPC, MenSCs, and WJ-MSCs expressed neuronal stemness protein markers. Effectively, iPSC-derived NPCs (Figure 3A,B), MenSCs (Figure 3C,D), and WJ-MSCs (Figure 3E,F) expressed SOX2 and Nestin (Figure 3G,H).

### 2.3. WT and PSEN 1 E280A MenSCs and WJ-MSCs Can Transdifferentiate into ChLNs and CSs Similarly to iPSC-Derived NPC

We further cultured WT and mutant iPSC-induced NPC (Figure 4A,B), MenSCs (Figure 4C,D), and WJ-MSCs (Figure 4E,F) in *Cholinergic-N-Run* (Figure 2A and Figure 4G) to obtain cholinergic-like neurons (ChLNs, Figure 4H–M), and assessed them by flow cytometry analysis (Figure 4N–P) and immunofluorescence microscopy (Figure 4Q–Y). The highest percentage of ChLNs was obtained from both WT and mutant WJ-MSCs (Figure 4P), whereas iPSCs (Figure 4N) and MenSCs (Figure 4O) showed no difference. Similar observations were recorded by immunofluorescent microscopy (Figure 4Q–Y).

When the NPC (obtained at day 27) and MSCs were exposed to *Fast-N-Spheres V2* medium (Figure 2B and Figure 5A), cerebral spheroids (CSs) were clearly identified (Figure 5B–G) in a process that lasted eight additional days and only 11 days for MenSCs and for WJ-MSCs. To further verify the cholinergic phenotype in the CSs, WT and mutant MenSCs-, WJ-MSCs-, and NPC-derived cholinergic cells were assessed for expression of neuronal markers. Figure 5 shows co-expression of the neuronal marker ChAT/VAChT/β III tubulin detected in NPCs-derived CSs (Figure 5H,I), MenSCs-derived CSs (Figure 5J,K), and WJ-MSCs-derived CSs (Figure 5L,M), showing that mutant cells express less β III tubulin (Figure 5N) and ChAT (Figure 5P) compared to WT, but no difference was found in the expression of VAChT (Figure 5O).

### 2.4. PSEN 1 E280A ChLNs Derived from MenSCs and WJ-MSCs Display Typical iAPPβf, p-Tau, and Oxidative Stress (oxDJ-1) Markers as Mutant NPC-Derived ChLNs

Based on the above observations, NPC-, MenSCs-, and WJ-MSCs-derived ChLNs were further evaluated for early accumulation of iAPPβf, p-Tau, and oxidized DJ-1, as evidence of OS. As shown in Figure 6, all three WT cellular sources show no detectable accumulation of iAPPβf (Figure 6A′,B,C), oxDJ-1 (e.g., Figure 6A″,B,C), nor p-Tau (Figure 6G′,H,I). In contrast, mutant ChLNs derived from MenSCs, WJ-MSCs, and NPC express abundant accumulation of iAPPβf (Figure 6D′,E,F), ox DJ-1 (Figure 6D″,E,F), and p-Tau (Figure 6J′,K,L). Statistical analysis revealed significant differences between WT and PSEN 1 E280A in production of iAPPβf (Figure 6M), ox DJ-1 (Figure 6N), and p-TAU (Figure 6O) compared to WT. PSEN 1 E280A WJ-MSCs-derived ChLNs generated the highest amount of iAPPβf, but MenSCs-derived ChLNs showed the highest amount of ox DJ-1. No statistically significant differences were found in p-Tau from the ChLNs. Similar data were obtained by flow cytometry analysis (Figure 6P–U).

Further analysis shows that the ΔΨ_m_ in ChLNs derived from WT PSEN 1 NPCs (Figure 7A,A′), MenSCs (Figure 7D,D′), and WJ-MSCs (Figure 7G,G′) was unaffected when compared to PSEN 1 E280A ChLNs (Figure 7B,B′,E,E′,H,H′), which presented a marked loss of ΔΨ_m_ (Figure 7C,F,I). When the generation of ROS (e.g., H_2_O_2_) was examined according to dichlorofluorescein (DCF)-positive cells, there were significantly more DCF+ cells in mutant ChLNs from the three cellular sources than WT ChLNs (Figure 7C,F,I). The loss of ΔΨ_m_ and generation of ROS (H_2_O_2_) were found to be similar in mutant ChLNs from any source.

### 2.5. PSEN 1 E280A ChLNs Derived from MenSCs, and WJ-MSCs Show Cell Death Markers of Apoptosis as Mutant NPC-Derived ChLNs

Next, we evaluated whether mutant ChLNs express apoptosis markers. Figure 8 shows that the protein PUMA was constitutively expressed at low levels in WT ChLNs derived from NPC (Figure 8A,A′), MenSCs (Figure 8B), and WJ-MSCs (Figure 8C,M). In contrast, PUMA showed a statistically significant increase in PSEN 1 E280A ChLNs from the three cellular sources (Figure 8D,D′,E,F) with almost similar strength (Figure 8M). Although almost no basal expression of p-JUN (Figure 8A,A″,B,C,N), TP53 (Figure 8G,G′,H,I,O), and CASP3 (Figure 8G,G″,H,I,P) was detected in WT ChLNs, these proteins showed a statistically significant rise in PSEN 1 E280A ChLNs from NPC (Figure 8D,D″,J,J′,J″,N–P), MenSCs (Figure 8E,K,N–P), and WJ-MSCs (Figure 8F,L,N–P). Similar data were obtained by flow cytometry analysis (Figure 8Q–V).

### 2.6. PSEN 1 E280A ChLNs Derived from NPC, MenSC and WJ-MSCs Do Not Respond to Acetylcholine (ACh) Stimuli

To evaluate whether ChLNs derived from NPCs, MenSCs, and WJ-MSCs were responsive to neurotransmitter stimuli as an assessment of cholinergic neuronal functionality, the WT and mutant ChLNs were exposed to acetylcholine (ACh), and Ca^2+^ influx was recorded in fluorescent microscopy. As shown in Figure 9, ACh induced a comparable transient elevation of intracellular Ca^2+^ in WT ChLNs derived from NPC (Figure 9A), MenSCs (Figure 9D), and WJ-MSCs (Figure 9G) with an average fluorescence change (DF/F) of 3.8 ± 0.6-fold, and a mean duration of 40 ± 10 s (n = 20 ChLN cells imaged, N = 3 dishes/each cell source) according to cytoplasmic Ca^2+^ response to Fluo-3-mediated imaging (Figure 9C,F,I). Remarkably, PSEN 1 E280A ChLNs obtained from the three cellular sources almost did not respond to ACh (Figure 9B,C,E,F,H,I).

### 2.7. PSEN1 E280A ChLNs Derived from WJ-MSCs Secrete Higher Amount of Extracellular Aβ42 Than Mutant ChLNs Derived from NPC and MenSC

Measurement of secreted eAβ42 (expressed as the ratio eAβ42/eAβ40) is critical for early detection and the disease-modifying treatments necessary to combat AD. Therefore, to assess the amounts of eAβ42 and eAβ40 secreted by both WT and mutant ChLNs derived from NPC, MenSCs, and WJ-MSCs, supernatants from culture medium were evaluated by a solid-phase sandwich ELISA according to the standard procedure described in the Section 4. The amount of secreted eAβ40 was almost constant in both WT and mutant PSEN 1 E280A ChLNs derived from NPC (Figure 10A), MenSCs (Figure 10B), and WJ-MSC (Figure 10C) ChLNs. However, secreted eAβ42 was significantly higher in the mutant ChLNs when compared to WT ChLNs from all three sources (Figure 10D–F). Interestingly, PSEN 1 E280A ChLNs derived from WJ-MSCs secreted greater amounts of eAβ42 (Figure 10F, 133 ± 17 pg/mL) than mutant NPC (Figure 10D, 53 ± 5 pg/mL), or mutant MenSCs (Figure 10E, 31 ± 1 pg/mL). As a result, the ratio Aβ42/40 was consistently higher in PSEN 1 E280A ChLNs derived from WJ-MSCs (Figure 10I, Aβ42/40 = 6) than the other two cellular sources (Figure 10G, Aβ42/40 = 1.6; and Figure 10H, Aβ42/40 = 1.3). In other words, PSEN 1 E280A WJ-MSCs-derived ChLNs secreted 3.75- and 4.62-folds of Aβ42 compared to mutant ChLNs derived from NPC and MenSCs, respectively.

### 2.8. PSEN 1 1E280A Cerebral Spheroids (CSs) Derived from MenSCs, and WJ-MSCs Display Typical iAPPβf, p-Tau and Oxidative Stress (oxDJ-1) Markers as Mutant NPCs-Derived CSs

Next, we wondered whether the accumulation of iAPPβ fragments, p-Tau, and oxidation of DJ-1 detected in 2D also occurred in 3D (CSs) structures. Figure 11 shows that no OS, iAPPβf or p-Tau markers were present in WT CSs from NPC (Figure 11A,A′,A″,B,C,G–I,M–O). In contrast, PSEN 1 E280A NPC-, MenSC-, and WJ-MSCs-derived CSs displayed the typical iAPPβf (Figure 11D,D′,E,F), OS (Figure 11D,D″,E,F), and p-Tau (Figure 11J,J′,K,L) markers, albeit with different levels of intensity (Figure 11M–O). While mutant WJ-MSCS-derived CSs generated higher levels of iAPPβf i.e., in terms of MFI signal, compared to mutant NPC-derived CSs (Figure 11M), mutant NPC-derived CSs showed significantly higher levels of oxidized DJ-1 protein (Figure 11N) compared to either mutant MenSCs-, or mutant WJ-MSCs-derived CSs. Yet, mutant WJ-MSC-derived CSs showed higher levels of p-Tau compared to mutant NPC-derived CSs (Figure 11O).

### 2.9. PSEN 1 1E280A CSs Derived from MenSCs and WJ-MSCs Show Markers of Apoptosis as Mutant NPC-Derived CSs

Based on the above observations, we wanted to confirm that CSs express apoptotic markers. To this aim, phosphorylated JUN, PUMA, TP53, and CASP3 were assessed in both WT and PSEN 1 E280A CSs. As expected, WT NPCs-, MenSCs-, and WJ-MSCs-derived CSs show neither PUMA (Figure 12A,A′,B,C), nor p-JUN (Figure 12A,A″,B,C), nor TP53 (Figure 12G,G′,H,I), nor CASP3 (Figure 12G,G″,H,I) markers (Figure 12M–P). The markers were visible in mutant CSs. Effectively, PSEN 1 E280A NPCs-, MenSCs-, and WJ-MSCs-derived CSs displayed a statistically significant increase in PUMA (Figure 12D,D′,E,F), p-JUN (Figure 12D,D″,E,F), TP53 (Figure 12G,J′,K,L), and CASP3 (Figure 12J,J″,K,L) markers, albeit without following a logical order of expression or activation (Figure 12M–P). While p-JUN was significantly increased in mutant CSs derived from NPC (Figure 12M), activated PUMA was significantly higher in mutant MenSCs (Figure 12N). The apoptotic marker TP53 was significantly increased in both mutant CSs derived from MenSCs and WJ-MSCs (Figure 12O). Like TP53, the executer protein CASP3 was significantly active in both mutant CSs derived from MenSCs and WJ-MSCs (Figure 12P).

## 3. Discussion

Given their resemblance to embryonic stem cells (ESCs), human iPSCs [31] have been instrumental for AD modeling in vitro [32] to understand the underlying mechanisms of the disease (e.g., [33]), test potential drugs [34], and develop personalized therapies [35]. Here, we obtained iPSCs from a WT PSEN 1 subject and from a patient with FAD bearing the mutation PSEN 1 E280A and used both as reference 2D and 3D tissue cultures for revealing the natural neuropathology of FAD and for comparative purposes. Like iPSCs, we established that multipotent MenSCs and WJ-MSCs: (i) expressed the pluripotent-associated markers OCT4, SOX2, NANOG, and KLF4; (ii) differentiated into cells of ectoderm and mesoderm germ layers; (iii) highly expressed neuronal stem marker nestin, and neural stem and progenitor cell marker SOX2; (iv) transdifferentiated into ChLNs, which expressed the cholinergic marker ChAT/VAChT (56 ± 6 & 79 ± 4%); and (vi) transdifferentiated into CSs. Taken together, these observations suggest that MenSCs and WJ-MSCs might be developmentally equivalent to iPSCs. Interestingly, the protocol to obtain ChLNs and CSs from MenSCs and WJ-MSCs lasts 11 days, whereas the protocol for obtaining neurons from iPSCs takes no less than 35 days. Therefore, using both *Cholinergic-N-Run* [29] and *Fast-N-Spheres V2* [30] medium protocols, at least 24 days are free from laboratory work (i.e., a 68% reduction in labor time). It was concluded that under the present conditions, both culture media are highly efficient as inducers of ChLNs. Since cholinergic neurons that form the nucleus basalis of Meynert are the most vulnerable to AD pathology [36], and therefore more severely lost, we considered that most of the 2D and 3D ChLNs in culture are reasonably homologous to ChNs in vivo.

In the present work, we report for the first time that iPSC::NPC-derived (planar or 2D culture) PSEN 1 E280A ChLNs and (3D culture) CSs displayed the neuropathological markers of AD, such as iAPPβf, and the secretion of high amounts of eAβ and p-TAU. Similar observations were recorded in 2D and 3D PSEN 1 E280A ChLNs derived from MenSCs (this work) and WJ-MSCs [29,30]. These observations suggest that the abnormal intracellular accumulation of Aβ seen in PSEN 1 E280A ChLNs and CSs is the earliest pathological event of a continuous process from an initial accumulation of iAPPβf to TAU phosphorylation [37] to the well-established extracellular Aβ aggregation, culminating in the formation of amyloid plaques [38] and cholinergic cell death. It was concluded that iAPPβf plays an important role in the genesis of PSEN 1 FAD [5]. Although several drugs that remove eAβ have failed to demonstrate clinical efficacy, including PSEN 1 E280A-derived Aβ [39], our in vitro data suggest that other alternative therapies against iAβ [40] or Tau protein [41] might be tested. Interestingly, we found oxidation of sensor protein DJ-1 (Cys^106^-SO_3_) concomitantly with iAPPβf accumulation in PSEN1 E280A iPSCs::NPC-derived ChLNs and CSs. Like mutant iPSCs::NPC-derived neurons, mutant ChLNs derived from MenSCs and WJ-MSCs also consistently showed DJ-1 (Cys^106^-SO_3_) together with iAPPβf. However, how exactly these two phenomena relate to each other is not yet established. One possible explanation is that iAPPβf directly or indirectly causes mitochondrial electron transport disruption [42,43], thereby bursting into ROS production (e.g., H_2_O_2_), which in turn might specifically oxidize DJ-1 [44]. In accordance with this assumption, we found that mutant ChLNs and CSs generated ROS (H_2_O_2_), and induced loss of ΔΨ_m_ concomitantly with Cys^106^-SO_3_. Whatever the connection might be, we consistently found that neurons expressed amyloidogenic Aβ, mitochondrial damage, ROS generation, and OS in planar ChLNs and (3D) CSs derived from the three biological sources (iPSCs::NPC, MenSCs, WJ-MSCs). Given that DJ-1 is not only an important stress sensor protein, but also modulates several signaling cellular pathways [45], it becomes a potential biomarker and therapeutic target in AD [46,47].

Interestingly, PSEN 1 E280A ChLNs derived from iPSCs::NPC, MenSCs, and WJ-MSCs reliably showed up-regulation of JUN, TP53, PUMA, and activation of CASP3, all involved in intrinsic apoptosis [48]. Accordingly, the iAPPβf generated H_2_O_2_ either oxidizes DJ-1 or can function as a second messenger [49], thereby activating other redox signaling proteins (e.g., apoptosis signal-regulating kinase 1 [50,51]), in turn activating the c-Jun N-terminal Kinase (JNK) pathway [52]. Remarkably, JNK can activate the transcription factor JUN [52], the transcription factor TP53 [53], and phosphorylate the protein Tau [54]. Therefore, JNK might also be a potential therapeutic target for AD [55]. Remarkably, both JUN [56] and TP53 [57,58] transactivate BH3-only protein PUMA, which in turn directly or indirectly induces loss of ΔΨ_m_ through the Bcl-2 pro-apoptotic protein Bax [59]. This last event on mitochondria leads to the release of apoptogenic proteins [60] and the activation of the cellular end executer protein CASP3, which is responsible for neuronal dismantling. Taken together, these results comply with the notion that iAPPβf induces a cascade of events involving OS-signaling, mitochondrial depolarization, p-Tau, and apoptosis [29,61,62] in ChLNs derived from the three biological sources examined. Interestingly, it has been shown that PSEN1 E280A ChLNs derived from WJ-MSCs did not respond to ACh-induced Ca^2+^ influx, most probably due to eAβ42 interaction with nicotinic acetylcholine receptors [63]. We report for the first time that iPSCs::NPC- and MenSCs-derived PSEN 1 E280A ChLNs were resilient to ACh-induced Ca^2+^ influx. Together, these results suggest that eAβ42 affects neuronal Ca^2+^ flux in PSEN 1 E280A independently of the cellular source tested.

Despite the use of iPSCs, these cells present several disadvantages. For instance, reprogramming highly depends on the efficient delivery and the suitable expression of Yamanaka factors, i.e., OSKM, into specific cell types (e.g., fibroblasts, cord blood CD133+ cells, peripheral blood mononuclear cells). Furthermore, the protocol(s) for obtaining reprogrammed cells is rather a slow and vulnerable process that may be affected by several factors (e.g., biopsy, cell type, particular culture conditions, long culture period) that hinder the efficiency, reproducibility, and quality of the resulting iPSCs (e.g., [64,65,66]). Therefore, replacement of iPSCs with a more manageable, cost-effective, time-saving biologic source that offers similarities to iPSCs is highly desirable.

In conclusion, we have demonstrated that PSEN 1 E280A ChLNs and cholinergic CSs derived from MenSCs, WJ-MSCs, and iPSCs can reliably reproduce the neuropathology of FAD in vitro, and therefore, they are cellularly and biochemically equivalent. Accordingly, PSEN 1 E280A iPSCs can be interchangeable with PSEN 1 E280A MenSCs and PSEN 1 E280A WJ-MSCs. Furthermore, they expressed the typical cellular hallmarks of AD, i.e., eAβ42 and p-TAU. Additionally, the presence of iAβ, oxidative markers DJ-1 Cys^106^-SO_3_ and p-JUN, loss of ΔΨ_m_, apoptosis markers TP53, PUMA, and CASP3, and dysfunctional ACh-induced Ca^2+^ influx were observed in mutant ChLNs and CSs in 11 days, whereas at least 35 days of culture are necessary to reproduce AD markers from mutant iPSCs::NPCs. This labor time-gap in favor of MenSCs and WJ-MSCs makes these biological sources much more attractive not only for modeling FAD but also for speeding up drug discoveries (e.g., antioxidant compounds [67]). However, to fully validate our findings, dissecting transcriptomic signatures of cholinergic neuronal differentiation using PSEN 1 E280A iPSCs (as reference tissue [68]), MenSCs and WJ-MSCs is warranted. Whatever the cause, cholinergic degeneration remains one of the earliest, most severe, and most consistent cellular changes in AD. Therefore, studying the cellular and molecular changes in cholinergic neurons may provide clues to the pathogenesis and treatment of this disorder [69]. Moreover, MSC-based therapy might be a promising alternative for the treatment of AD [70,71].

## 4. Materials and Methods

### 4.1. Human Menstrual Stromal Cells (MenSCs) and Wharton Jelly-Mesenchymal Stromal Cells (WJ-MSCs)

The menstrual blood (MenB) samples were collected from one healthy female and one female carrier of the mutation PSEN1 E280A aged 30 years (Tissue Bank Code, TBC#69308) and 25 years (TBC#04335) according to ref. [28]. The WT (TBC# WJ-MSC-15) and PSEN1 E280A (TBC# WJ-MSC-12) were collected as described previously [29]. Donors signed an informed consent accepted by the Ethics Committee of the Sede de Investigación Universitaria-SIU-, Act#2020-10854, University of Antioquia, Medellín, Colombia.

### 4.2. Cholinergic-like Neuron (ChLN) Differentiation

ChLN differentiation was performed according to ref. [28]. Briefly, WT and mutant MenSCs and WJ-MSCs were seeded at 1–1.5 × 10^4^ cells/cm^2^ in laminin-treated culture plates for 24 h in regular culture medium. Then, the medium was removed, and cells were incubated in cholinergic differentiation medium (*Cholinergic-N-Run* medium, hereafter *Ch-N-Rm*) at 37 °C for 7 days, and then the medium was replaced by minimal culture medium (mCM) for 4 days.

### 4.3. Human Induced Pluripotent Stem Cell (iPSC) Lines Culture and Differentiation

Reprogramming of human primary skin fibroblasts from 2 adult donors (PSEN1 WT: Ctrl-4 and E280A: p106) was performed using a single, multicistronic lentiviral vector encoding OCT4, SOX2, KLF4, and MYC according to ref. [31] in Dr. Kosik’s laboratory. Donors provided written informed consent in accordance with the Medical Ethical Committee of the University of Antioquia. The iPSC cells were thawed and cultivated on VTN-N (Cat# A14700, Thermo Fisher Scientific Inc., Santa Fe, NM, USA)-coated 6-well plates to form colonies in Drs. Velez-Jimenez’s laboratory. Human iPSC cells were mechanically detached from the VTN-N surface. Embryoid bodies (EBs) were generated by transferring iPSCs to non-adherent plates in E6 medium at 37 °C in 5% CO_2_. After 7 days, EBs were transferred to a non-adherent plate, and E6 medium (Cat# A1516401, GIBCO, Santa Fe, NM, USA) was supplemented with 10 ng/mL bFGF (Cat# F0291-25UG, Sigma-Aldrich, St. Louis, MO, USA). After 2 days, the floating structures were dissociated by trituration and transferred to a VTN-N-treated dish. For generation of neural progenitor cells (NPCs), EBs were cultured in NPC medium (Neurobasal medium, Cat# 21103049, GIBCO, Santa Fe, NM, USA; 1% N2 supplement, Cat# 17502048, GIBCO, Santa Fe, NM, USA; 2% B27 supplement, Cat# 17504044, GIBCO, Santa Fe, NM, USA), 20 ng/mL epidermal growth factor (Cat# E5036, Sigma-Aldrich, St. Louis, MO, USA), 1 μg/mL heparin sodium salt (Cat# 375095, Sigma-Aldrich, St. Louis, MO, USA), 1 ng/ml bFGF (Cat# F0291-25UG, Sigma-Aldrich, St. Louis, MO, USA), 1× β-mercaptoethanol (Cat# M3148-100ML, Sigma-Aldrich, St. Louis, MO, USA) and 1% penicillin/streptomycin (Cat# 15140122, GIBCO, Santa Fe, NM, USA) for 18 days. NPC were seeded at 3 × 10^4^ cells/cm^2^ in 24-well culture flasks for 24 h in NPC culture medium. Then, the medium was removed, and cells were incubated in cholinergic differentiation medium (*Cholinergic-N-Run*) at 37 °C for 2 days [19]. After cholinergic induction, the medium was replaced and refreshed with neural medium (NM), composed by neurobasal medium supplemented with 1× N2 and 1% penicillin/streptomycin.

### 4.4. Generation of Cerebral Spheroids (CSs)

Cerebral spheroids were obtained by differentiation of WT or PSEN1 E280A from iPSC-derived NPC, MenSCs, and WJ-MSCs, as previously described in ref. [30]. Briefly, WT and mutant NPC, MenSCs, or WJ-MSCs were cultured in a brand-new medium called *Fast-N-Spheres V2* plus Corning Matrigel^®^ (Cat# 356232, Thermo Fisher Scientific Inc., Santa Fe, NM, USA) and 1% fetal bovine serum (FBS; Cat# CVFSVF00-01, Eurobio Scientific, Les Ulis, France). Cultures were constantly shacked for 2 days. Then, CSs medium was replaced by neuronal medium (NM) and left under incubation for 6 or 11 days, depending on tissue resources.

Evaluation of pluripotent-, and neuronal-associated markers Initially, iPSCs, MenSCs, and WJ-MSCs were characterized for the pluripotent-associated markers OCT4 (Cat# A13998, Invitrogen, Waltham, MA, USA), SOX2 (Cat# PA1-094, Thermo Fisher Scientific Inc., Santa Fe, NM, USA), NANOG (Cat# AB9220, Millipore, Burlington, MA, USA) and KLF4 (Cat# MA5-15695, Invitrogen, Waltham, MA, USA). Then, iPSCs were differentiated into NPC and subsequently differentiated into cholinergic cells, or CSs. Similarly, MenSCs and WJ-MSCs were transdifferentiated into ChLNs or cerebral spheroids. Finally, cells, or CSs, were used to evaluate neuronal markers and AD-associated pathological proteins.

### 4.5. Immunofluorescence Analysis

The immunofluorescence analysis of the different neuropathological markers related to FAD was performed according to ref. [29]. Briefly, the neuronal cells, or CSs incubated under different conditions were fixed with cold ethanol (−20 °C; Cat# 459836-500ML, Sigma-Aldrich, St. Louis, MO, USA) for 20 min. Then, cells were washed three times over 5 min with PBS, followed by Triton X-100 (0.1%; Cat# 93443, Sigma-Aldrich, St. Louis, MO, USA) permeabilization and 10% bovine serum albumin (BSA; Cat# A9418, Sigma-Aldrich, St. Louis, MO, USA) blockage for 30 min at room temperature. Cells were incubated overnight with primary antibodies against the pluripotency transcription factors OCT4 (1:500), SOX-2 (1:500), NANOG (1:500), and KLF4 (1:500) and the neuronal marker Nestin (1:500; cat# MA1-5840, Invitrogen, Waltham, MA, USA). Neuronal precursor markers such as β-3 tubulin, GFAP, ChAT, the pathological/apoptosis associated proteins APP751 and/or protein amyloid β1-42, total TAU, phospho-TAU, PUMA, p53, caspase-3, and oxidative stress markers DJ-1 and phospho-c-Jun were used according to ref. [29]. Finally, cells were washed (three times over 5 min with PBS) and incubated with secondary fluorescent antibodies DyLight 488 and 594 (cat# DI 1094, cat# DI 3088, and cat# DI 2488, Vector Laboratories, Newark, NJ, USA), according to ref. [29].

### 4.6. Evaluation of Intracellular Hydrogen Peroxide (H_2_O_2_) by Fluorescence Microscopy

The levels of intracellular H_2_O_2_, were determined according to ref. [29]. Briefly, cells were left in neural medium (NM) for 0 and 4 days or in minimal culture media (mCM) for 4 days. Then, the cells (5 × 10^3^) were incubated with 2′,7′-dichlorofluorescein diacetate reagent (5 μM, DCFH_2_-DA; Cat# D399, Invitrogen, Waltham, MA, USA) for 30 min at 37 °C in the dark. Cells were then washed, and DCF fluorescence intensity was determined by analysis of fluorescence microscopy images. The assessment was repeated three times in independent experiments, blind to the experimenter.

### 4.7. Analysis of Mitochondrial Membrane Potential (ΔΨ_m_) by Fluorescence Microscopy

The ChLNs were left in NM for 0–4 days or in mCM for 4 days. Then, the cells (5 × 10^3^) were incubated with the passively diffusing and active mitochondria-accumulating dye deep red MitoTracker compound (20 nM, final concentration) for 20 min at RT in the dark (cat# M22426, Invitrogen, Waltham, MA, USA) according to ref. [29]. The assessment was repeated three times in independent experiments.

### 4.8. Intracellular Calcium Imaging

Intracellular calcium (Ca^2+^) concentration changes evoked by cholinergic stimulation were assessed according to refs. [72,73] with minor modifications. Briefly, for the measurement, the fluorescent dye Fluo-3 (Fluo-3 AM; cat: F1242 Thermo Fisher Scientific, Santa Fe, NM, USA) was employed. Intracellular Ca^2+^ transients were evoked by acetylcholine (Ach; Cat# A2661, Sigma-Aldrich, St. Louis, MO, USA); 1 mM final concentration) at 4 days post differentiation. The amplitudes of the Ca^2+^-related fluorescence transients were expressed relative to the resting fluorescence (ΔF/F) and were calculated by the following formula: ΔF/F = (F_maximum_ − F_resting_)/(F_resting_ − F_background_). For the calculation of the fluorescence intensities, ImageJ was used. The assessment was repeated three times in independent experiments, blind to the experimenter.

### 4.9. Measurement of Aβ 1–40 and Aβ 1–42 Peptides in Culture Medium

The levels of Aβ 1–40 and Aβ 1–42 peptides were measured according to a previous report with minor modifications. Briefly, WT and PSEN1 E280A cells were left in NM or mCM for 4 days. Then, 100 μL of supernatants were collected, and the levels of secreted Aβ1–40 and Aβ1–42 peptides were determined by a solid-phase sandwich ELISA (Cat# 150496 and Cat# 574166, respectively, Abbexa, Cambridge, UK) following the manufacturer’s instructions. The assessment was repeated three times in independent experiments, blind to the experimenter.

### 4.10. Photomicrography and Image Analysis

Light microscopy photographs were taken using a Zeiss Axiostart 50 Fluorescence Microscope (Carl Zeiss, Gottingen, Germany) equipped with a Canon PowerShot G5 digital camera (Zeiss Wöhlk-Contact-Linsen, Gmb Schcönkirchen, Gottingen, Germany), and fluorescence microscopy photographs were taken using a Zeiss Axiovert A1 Fluorescence Microscope equipped with a Zeiss AxioCam Cm1 and (Zeiss Wöhlk-Contact-Linsfluoreen, Gmb Schcönkirchen, Gottingen, Germany) and Floid Cells Imaging Station microscope (Cat# 4471136, Life Technologies, Carlsbad, CA, USA). Fluorescence images were analyzed by ImageJ software (http://imagej.nih.gov/ij/, accesed on 7 May 2023). Mean fluorescence intensity (MFI) was obtained by normalizing total fluorescence to the number of nuclei.

### 4.11. Data Analysis

In this experimental design, two vials of each specimen/origin were thawed (PSEN1-WT and -E280A), cultured, and the cell suspension was pipetted at a standardized cellular density of 2.6 × 10^4^ cells/cm^2^ into different wells of a 24-well plate. Experiments were conducted in triplicate wells, according to ref. [29]. The data from individual replicate wells were averaged to yield a value of n = 1 for that experiment, and this was repeated on three occasions, blind to the experimenter and/or flow cytometer analyst, for a final value of n = 3 [74]. The data were analyzed according to ref. [29]. The statistical significance was determined by one-way analysis of variance (ANOVA) followed by Tukey’s post hoc comparison calculated with GraphPad Prism 5.0 software (https://www.graphpad.com/; accessed on 5 February 2023). Differences between groups were only deemed significant when a *p*-value of <0.05 (*), <0.001 (**) and <0.001 (***). All data are illustrated as the mean ± S.D.

## Figures and Tables

**Figure 1 ijms-24-08957-f001:**
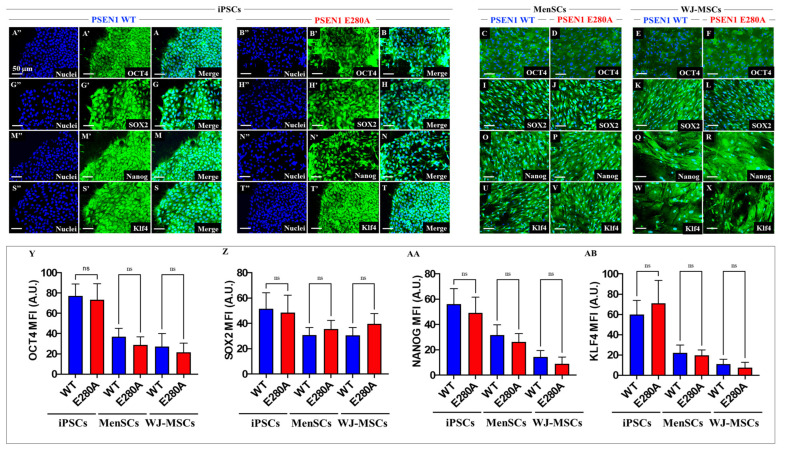
Determination of pluripotency markers by immunofluorescence. Nuclear colocalization of OCT4 in WT (**A**–**A”**) and PSEN1 E280A (**B**–**B″**) iPSCs, in WT (**C**) and PSEN1 E280A MenSCs (**D**), and in WT (**E**) and PSEN1E280A WJ-MSCs (**F**). Colocalization of SOX2 in WT (**G**–**G″**) and PSEN 1 E280A iPSCs (**H**–**H″**), in WT (**I**) and PSEN 1 E280A MenSCs (**J**), and in WT (**K**) and PSEN 1E280A WJ-MSCs (**L**). Colocalization of Nanog in WT (**M**–**M″**) and PSEN 1 E280A iPSCs (**N**–**N″**), in WT (**O**) and PSEN 1 E280A MenSCs (**P**), and in WT (**Q**) and PSEN 1 E280A WJ-MSCs (**R**). Colocalization of Klf4 in WT (**S**–**S″**) and PSEN 1 E280A iPSCs (**T**–**T″**), in WT (**U**) and PSEN 1 E280A MenSCs (**V**), and in WT (**W**) and PSEN 1 E280A WJ-MSCs (**X**). Quantitative data showing the nuclear mean fluorescence intensity for OCT4 (**Y**), SOX2 (**Z**), NANOG (**AA**), and KLF4 (**AB**). The figures represent one out of three independent experiments. The data are expressed as the mean ± SD; significant values were determined by one-way ANOVA with Tukey’s post hoc test; ns: not significant. Image magnification, 20×. Representative immunocytochemistry images of mesoderm germ layer stained for Vimentin (**AC**–**AH**), ectoderm stained for ectoderm stained for Nestin (**AI**–**AN**), and endoderm stained for CXCR4 (**AO**–**AT**) in WT and PSEN 1 E280 A mutation from iPSCs, MenScs, and WJ-MSCs. Nuclei are stained with Hoechst (blue). Scale bars 25 μm.

**Figure 2 ijms-24-08957-f002:**
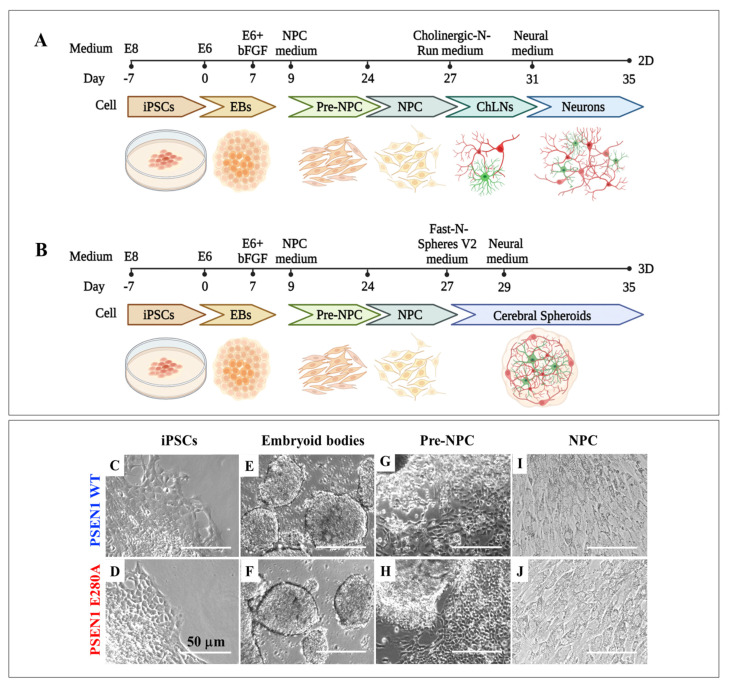
Schematic representation of the iPSCs differentiation protocol: (**A**,**B**) graphic timeline of iPSC-derived NPC, cholinergic neurons (ChLNs), or cerebral spheroids (CSs); (**C**,**D**) iPSC colony morphology from WT and PSEN1 E280A cells; (**E**,**F**) embryoid body (EB) morphology from WT and PSEN1 E280A cells; (**G**,**H**) pre-NPC morphology from WT and PSEN1 E280A cells; and (**I**,**J**) NPC morphology from WT and PSEN1 E280A cells.

**Figure 3 ijms-24-08957-f003:**
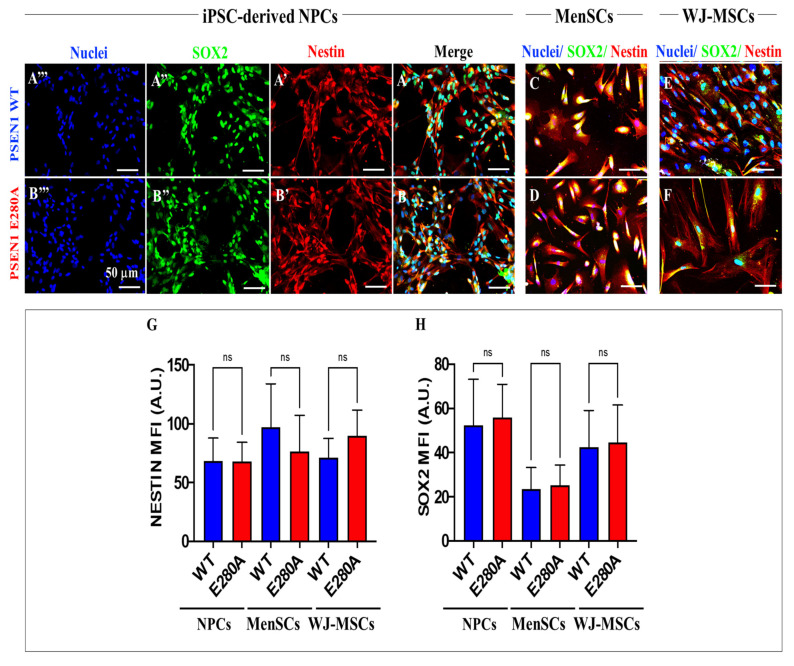
Determination of NPC markers by immunofluorescence. iPSCs were differentiated into NPCs as described in Section 4. Then, Nestin (**A′**,**B′**), SOX2 (**A″**,**B″**) proteins, and nuclei (**A‴**,**B‴**) were simultaneously identified in WT (**A**) and PSEN1 E280A (**B**) cells. MenSCs were cultured in regular MSC medium, then cells were stained to identify Nestin, SOX2 proteins, and nuclei in WT (**C**) and PSEN1 E280A (**D**) cells. WJ-MSCs were cultured in regular MSC medium, then cells were stained to identify Nestin, SOX2 proteins, and nuclei in WT (**E**) and PSEN 1 E280A (**F**) cells. Quantitative data showing the mean fluorescence intensity for cytosolic Nestin (**G**), and SOX2 (**H**). The figures represent one out of three independent experiments. The data are expressed as the mean ± SD; significant values were determined by one-way ANOVA with Tukey’s post hoc test; ns: not significant. Image magnification, 20×.

**Figure 4 ijms-24-08957-f004:**
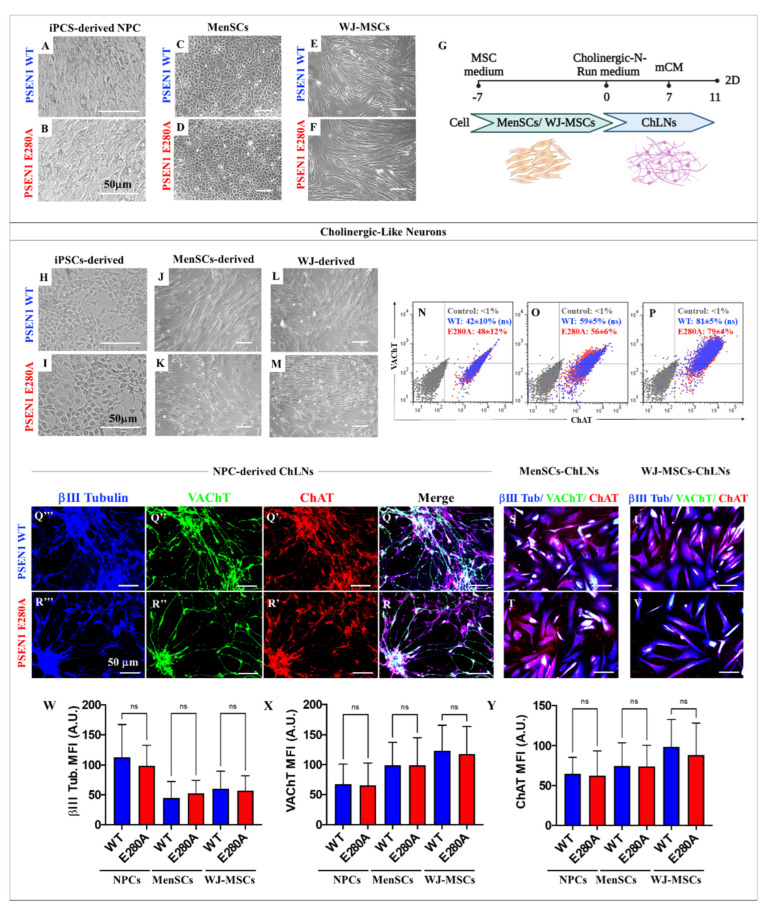
The iPSCs-, MenSCs-, and WJ-MSCs-derived cholinergic-like neurons (ChLNs): (**A**–**F**) light microscopy pictures of WT (**A**) and PSEN 1 E280A iPSC-derived NPC (**B**); undifferentiated WT (**C**) and mutant MenSCs (**D**); and undifferentiated WT (**E**) and mutant WJ-MSCs (**F**). Graphic timeline of WT and PSEN 1 E280A MenSCs- and WJ-MSCs-derived cholinergic-like neurons (ChLNs) (**G**); light microscopy pictures of WT (**H**) and PSEN 1 E280A iPSCs::NPC-derived ChLNs (**I**); WT (**J**) and mutant MenSCs-derived ChLNs (**K**), and WT (**L**) and mutant WJ-MSCs-derived ChLNs (**M**). Flow cytometry analysis of WT and PSEN 1 E280A iPSCs::NPC- (**N**), MenSCs- (**O**), and WJ-MSCs-derived ChLNs (**P**) to identify VAChT and ChAT. The histograms represent 1 out of 3 independent experiments. The data are expressed as the mean ± SD; significant values were determined by one-way ANOVA with Tukey’s post hoc test; ns = not significant. Representative fluorescence microscopy pictures of WT (**Q**) and PSEN 1 E280A NPC-derived ChLNs (**R**), WT (**S**) and mutant MenSCs-derived ChLNs (**T**), and WT (**U**) and mutant WJ-MSCs-derived ChLNs (**V**) stained with antibodies against ChAT (**Q′**,**R′**,**S**–**V**), VAChT (**Q″**,**R″**,**S**–**V**), and β III Tubulin (**Q‴**,**R‴**,**S**–**V**). Quantitative data showing the mean fluorescence intensity for β III Tubulin (**W**), VAChT (**X**), and ChAT (**Y**). The figures represent 1 out of 3 independent experiments. The data are expressed as the mean ± SD; significant values were determined by one-way ANOVA with Tukey’s post hoc test; Image magnification, 20×.

**Figure 5 ijms-24-08957-f005:**
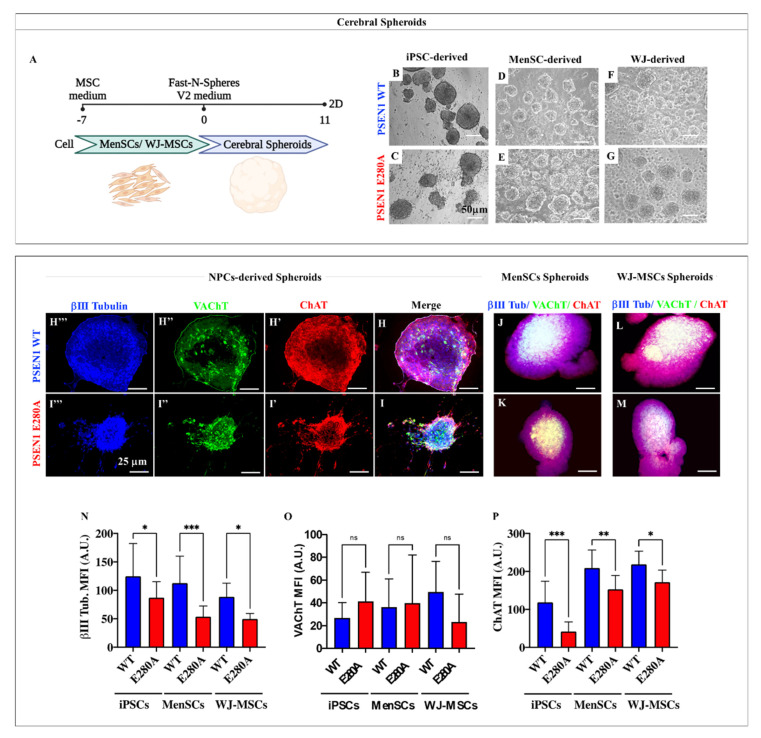
The iPSCs-, MenSCs-, and WJ-MSCs-derived cerebral spheroids (CSs) express cholinergic markers: (**A**) graphic timeline of WT and PSEN 1 E280A MenSCs- and WJ-MSCs-derived cerebral spheroids (CSs); (**B**–**G**) light microscopy pictures of WT (**B**) and PSEN 1 E280A iPCs::NPC-derived CSs (**C**), WT (**D**) and mutant MenSCs-derived CSs (**E**), and WT (**F**) and mutant WJ-MSCs-derived CSs (**F**). Representative fluorescence microscopy pictures of WT (**H**) and PSEN 1 E280A NPC-derived CSs (**I**), WT (**J**) and mutant MenSCs-derived CSs (**K**), and WT (**L**) and mutant WJ-MSCs-derived CSs (**M**) stained with antibodies against ChAT (**H′**,**I′**,**J**–**M**), VAChT (**H″**,**I″**,**J**–**M**), and β III tubulin (**H‴**,**I‴**,**J**–**M**). Quantitative data showing the mean fluorescence intensity for β III Tubulin (**N**), VAChT (**O**), and ChAT (**P**). The figures represent 1 out of 3 independent experiments. The data are expressed as the mean ± SD; significant values were determined by one-way ANOVA with Tukey’s post hoc test; * *p* < 0.05; ** *p* < 0.005; *** *p* < 0.001. ns: not significant. Image magnification, 20×.

**Figure 6 ijms-24-08957-f006:**
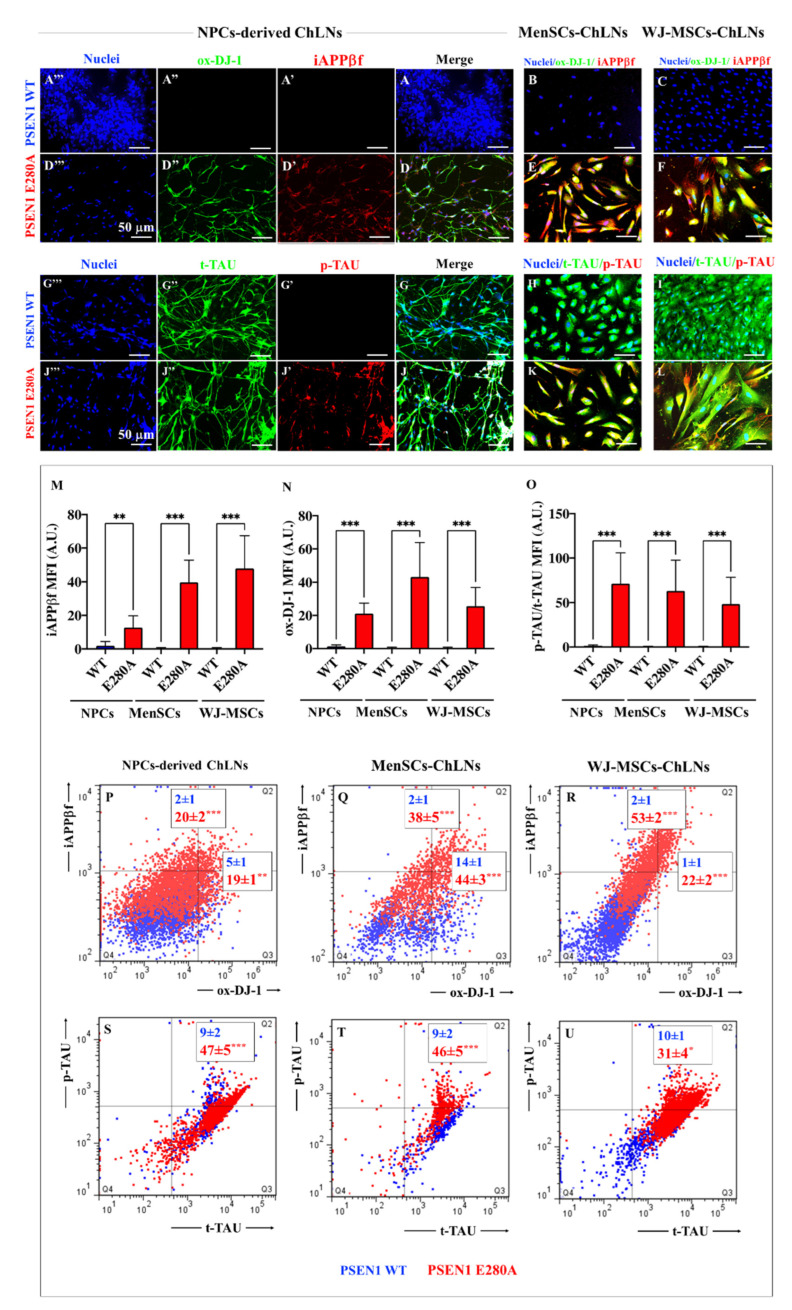
Determination of Alzheimer’s pathological proteins in ChLNs. Representative immunofluorescence microscopy of WT (**A**–**C**,**G**–**I**) and PSEN 1 E280A iPSCs::NPC-, MenSCs-, and JW-MSCs-derived ChLNs (**D**–**F**,**J**–**L**) stained to identify iAPPβf (**A′**,**D′**,**B**,**C**,**E**,**F**), ox-DJ-1 (**A″**,**D″**,**B**,**C**,**E**,**F**), p-TAU (**G′**,**J′**,**H**–**K**), t-Tau (**G″**,**J″**,**H**–**K**), and nuclei (**A‴**,**D‴**,**B**,**C**,**E**,**F**,**H**–**L**). Quantitative data showing the mean fluorescence intensity for iAPPβf (**M**), ox-DJ-1 (**N**), and the p-TAU/t-TAU ratio (**O**). The figures represent 1 out of 3 independent experiments. The data are expressed as the mean ± SD; significant values were determined by one-way ANOVA with Tukey’s post hoc test; * *p* < 0.05; ** *p* < 0.005; *** *p* < 0.001. Image magnification, 20×. Flow cytometry analysis of WT and PSEN 1 E280A iPSCs::NPC- (**P**,**S**), MenSCs- (**Q**,**T**), and WJ-MSCs-derived ChLNs (**R**,**U**) to identify iAPPβf (**P**–**R**), ox-DJ-1 (**P**–**R**), and p-Tau (**S**–**U**). The histograms represent 1 out of 3 independent experiments. The data are expressed as the mean ± SD; significant values were determined by one-way ANOVA with Tukey’s post hoc test; ** *p* < 0.005; *** *p* < 0.001.

**Figure 7 ijms-24-08957-f007:**
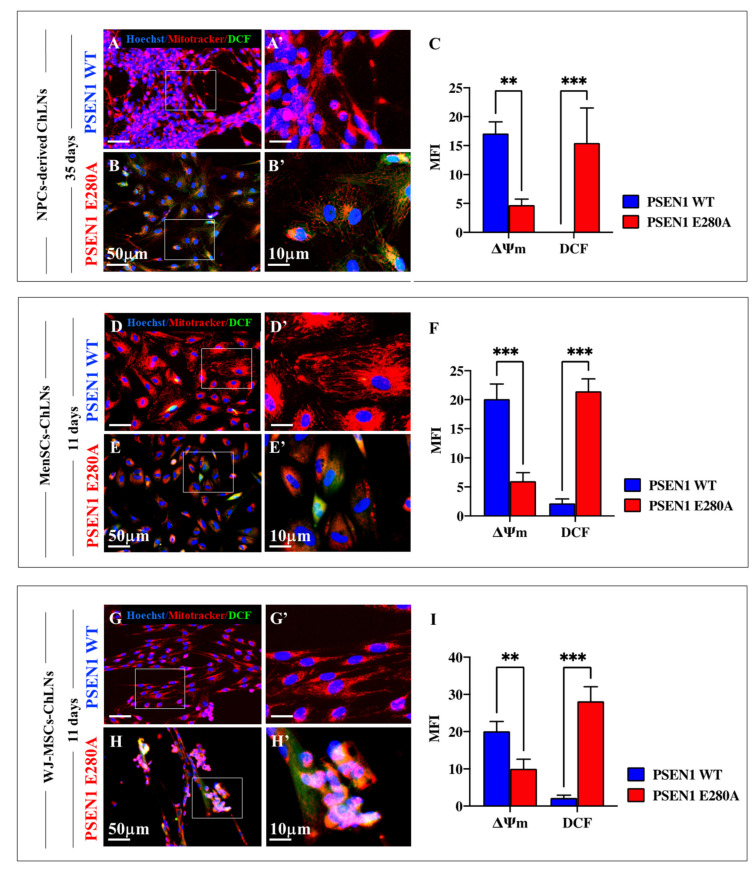
Evaluation of mitochondrial membrane potential (ΔΨ_m_) and ROS production in ChLNs. At day 35, WT and PSEN 1 E280A iPSC::NPC-derived ChLNs (**A**–**C**), and at day 11, WT and PSEN 1 E280A MenSCs (**D**–**F**), and WT and PSEN 1 E280A WJ-MSCs-derived ChLNs (**G**–**I**) were stained with Hoechst, red Mitotracker, and dichlorofluorescein to identify nuclei, ΔΨ_m_ and ROS production. Images were analyzed, and quantitative data were compared (**C**,**F**,**I**). The figures represent one out of three independent experiments. The data are expressed as the mean ± SD; significant values were determined by one-way ANOVA with Tukey’s post hoc test; ** *p* < 0.005; *** *p* < 0.001. Image magnification 20×. White square area is magnified inset (**A′**,**B′**,**D′**,**E′**,**G′**,**H′**), magnification 100×.

**Figure 8 ijms-24-08957-f008:**
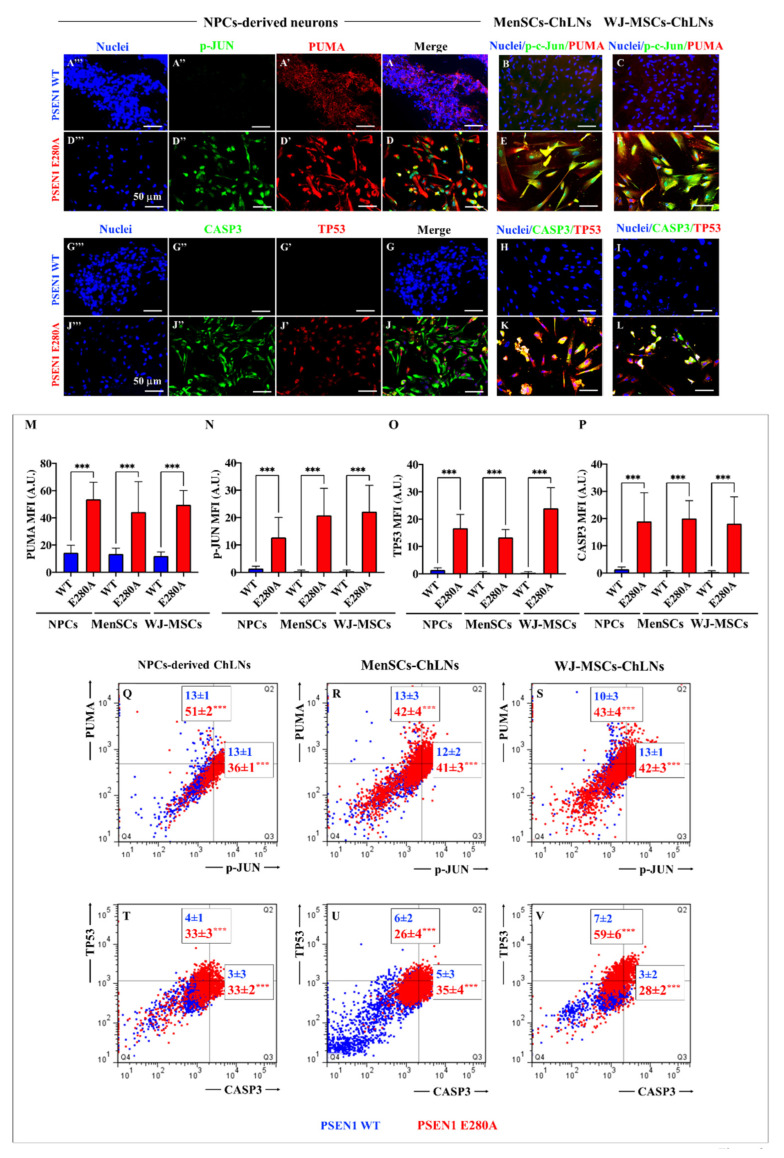
Determination of apoptosis-associated proteins in ChLNs. Representative immunofluorescence microscopy of WT (**A**–**C**,**G**–**I**) and PSEN 1 E280A iPSCs::NPC- (**D**,**J**), MenSCs- (**E**,**K**), and JW-MSCs-derived ChLNs (**F**,**L**) stained to identify PUMA (**A′**,**D′**,**B**,**C**,**E**,**F**), p-JUN (**A″**,**D″**,**B**,**C**,**E**,**F**), TP53 (**G′**,**J′**,**H**–**K**), CASP3 (**G″**,**J″**,**H**–**K**), and nuclei (**A‴**,**D‴**,**B**,**C**,**E**,**F**,**H**–**L**). Quantitative data showing the mean fluorescence intensity for PUMA (**M**), c-JUN (**N**), TP53 (**O**), and CASP3 (**P**). The figures represent 1 out of 3 independent experiments. The data are expressed as the mean ± SD; significant values were determined by one-way ANOVA with Tukey’s post hoc test; *** *p* < 0.001. Image magnification, 20×. Flow cytometry analysis of WT and PSEN 1 E280A iPSCs::NPC- (**Q**,**T**), MenSCs- (**R**,**U**), and WJ-MSCs-derived ChLNs (**S**,**V**) to identify PUMA (**Q**–**S**), p-JUN (**Q**–**S**), TP53 (**T**–**V**), and CASP3 (**T**–**V**). The histograms represent 1 out of 3 independent experiments. The data are expressed as the mean ± SD; significant values were determined by one-way ANOVA with Tukey’s post hoc test; *** *p* < 0.001.

**Figure 9 ijms-24-08957-f009:**
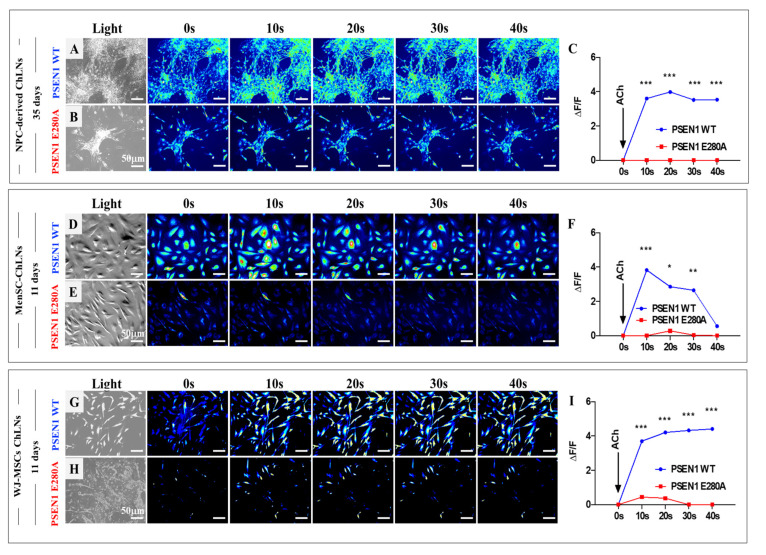
Evaluation of Acetylcholine (ACh) response. Time-lapse images (0, 10, 20, 30, and 40 s) of Ca^2+^ fluorescence in WT and E280A iPSC-derived neurons after 31 and 35 days (n = 3 dishes) as a response to ACh treatment (**A**,**B**,**D**,**E**). Time-lapse images (0, 10, 20, 30, and 40 s) of Ca^2+^ fluorescence in PSEN 1 WT and E280A MenSCs- and WJ-MSCs- derived ChLNs after 11 days (n = 3 dishes) as a response to ACh treatment (**G**,**H**) ACh was puffed into the culture at 0 s (arrow). Then, the Ca^2+^ fluorescence of cells was monitored at the indicated times. Color contrast indicates fluorescence intensity: dark blue < light blue < green < yellow < red. (**C**,**F**,**I**) Normalized mean fluorescence signal (ΔF/F) over time, indicating temporal cytoplasmic Ca^2+^ elevation in response to ACh treatment in PSEN 1 WT and E280A cells. The data are expressed as the mean ± SD; significant values were determined by one-way ANOVA with Tukey’s post hoc test; * *p* < 0.05; ** *p* < 0.005; *** *p* < 0.001. Image magnification, 20×.

**Figure 10 ijms-24-08957-f010:**
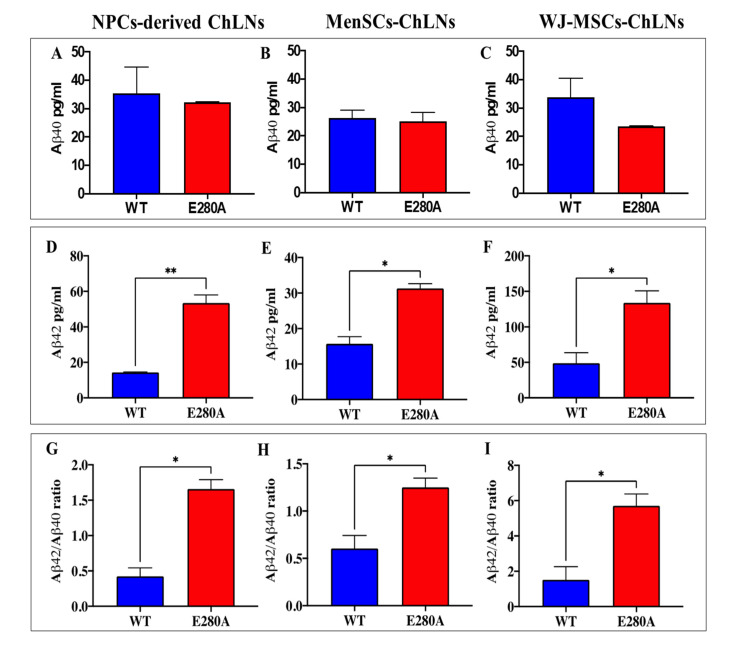
ELISA quantification of extracellular Aβ40 and Aβ42 peptides in supernatants from PSEN1 WT and PSEN 1 E280A cells. WT and PSEN1 E280A ChLNs cells were left in neural medium or minimal culture medium for 4 days. The levels of secreted Aβ1–40 and Aβ1–42 peptides were determined as described in the Section 4. The ELISA measurements of Aβ40 in the supernatant from WT and PSEN1 E280A cholinergic cells derived from iPSCs (**A**), MenSCs (**B**), and WJ-MSCs (**C**) cells at day 4. The ELISA measurements of Aβ42 in supernatant from WT and PSEN1E280A cholinergic cells derived from iPSCs (**D**), MenSCs (**E**), and WJ-MSCs (**F**) cells at day 4. The Aβ42 over Aβ40 ratio in PSEN1 E280A from iPSCs (**G**), MenSCs (**H**), and WJ-MSCs (**I**) compared with WT at day 4. The figures represent one out of three independent experiments. Significant values were determined by a one-way ANOVA with Tukey’s post hoc test. The data are presented as mean ± SD (* *p* < 0.05; ** *p* < 0.01).

**Figure 11 ijms-24-08957-f011:**
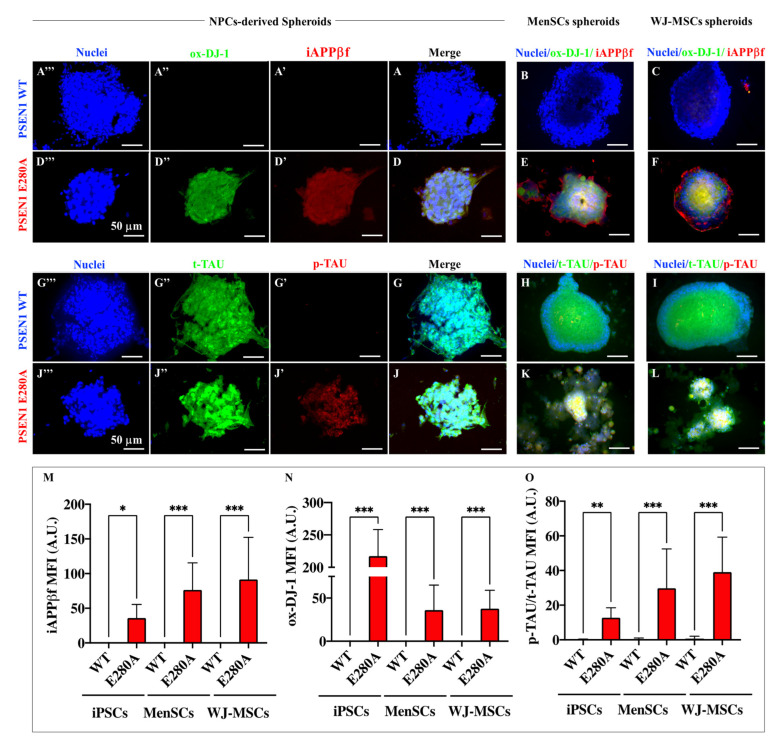
Determination of Alzheimer’s pathological proteins by immunofluorescence in cerebral spheroids (CSs). Representative immunofluorescence microscopy of WT (**A**–**C**,**G**–**I**) and PSEN 1 E280A iPSCs::NPC- (**D**,**J**), MenSCs- (**E**,**K**), and JW-MSCs-derived ChLNs (**F**,**L**) stained to identify iAPPβf (**A′**,**D′**,**B**,**C**,**E**,**F**), ox-DJ-1 (**A″**,**D″**,**B**,**C**,**E**,**F**), p-TAU (**G′**,**J′**,**H**–**K**), t-Tau (**G″**,**J″**,**H**–**K**), and nuclei (**A‴**,**D‴**,**B**,**C**,**E**,**F**,**H**–**L**). Quantitative data showing the mean fluorescence intensity for iAPPβf (**M**), ox-DJ-1 (**N**), and the p-TAU/t-TAU ratio (**O**). The figures represent one out of three independent experiments. The data are expressed as the mean ± SD; significant values were determined by one-way ANOVA with Tukey’s post hoc test; * *p* < 0.05; ** *p* < 0.005; *** *p* < 0.001. Image magnification, 20×.

**Figure 12 ijms-24-08957-f012:**
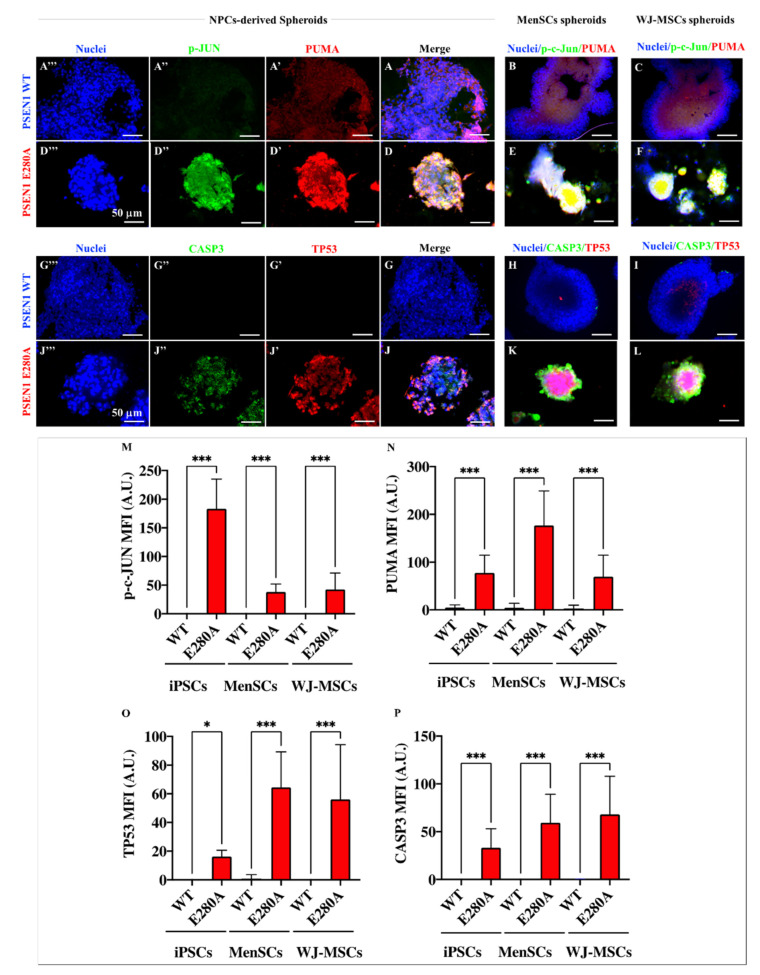
Determination of apoptosis-associated proteins by immunofluorescence in cerebral spheroids (CSs). Representative immunofluorescence microscopy of WT (**A**–**C**,**G**–**I**) and PSEN 1 E280A iPSCs::NPC- (**D**,**J**), MenSCs- (**E**,**K**), and JW-MSCs-derived ChLNs (**F**,**L**) stained to identify PUMA (**A′**,**D′**,**B**,**C**,**E**,**F**), p-c-JUN (**A″**,**D″**,**B**,**C**,**E**,**F**), TP53 (**G′**,**J′**,**H**–**K**), CASP3 (**G″**,**J″**,**H**–**K**), and nuclei (**A‴**,**D‴**,**B**,**C**,**E**,**F**,**H**–**L**). Quantitative data showing the mean fluorescence intensity for PUMA (**M**), c-p-JUN (**N**), TP53 (**O**), and CASP3 (**P**). The figures represent one out of three independent experiments. The data are expressed as the mean ± SD; significant values were determined by one-way ANOVA with Tukey’s post hoc test; * *p* < 0.05; *** *p* < 0.001. Image magnification, 20×.

## Data Availability

All relevant data are within the manuscript.

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
