# Peer review of "PSEN1 E280A Cholinergic-like Neurons and Cerebral Spheroids Derived from Mesenchymal Stromal Cells and from Induced Pluripotent Stem Cells Are Neuropathologically Equivalent"

_ijms, 2023, doi:10.3390/ijms24108957_

Round 1

Reviewer 1 Report

The manuscript entitled “Menstrual stromal cells- and Wharton Jelly’s mesenchymal stromal cells-derived PSEN1 E280A cholinergic-like neurons (ChLNs) and cerebral spheroids (CSs) are neuropathologically equivalent to PSEN1 E280A ChLNs/ CSs obtained from induced pluripotent stem cells-derived neural progenitor cells” is well written and complete in all aspect of the study. Here are my few minor comments to improve the quality of manuscript.

1.     Title is very large and difficult to follow. So, it should be shortened.

2.     Authors needs to discuss more about MSCs and MSC derived exosomes in the introduction as discussed in the studies below.

https://doi.org/10.1016/j.ejphar.2021.174657

https://doi.org/10.1016/j.lfs.2021.119465

Reviewer 2 Report

The manuscript entitled “ Menstrual stromal cells- and Wharton Jelly’s mesenchymal stromal cells-derived PSEN1 E280A cholinergic-like neurons (ChLNs) and cerebral spheroids (CSs) are neuropathologically equivalent to PSEN1 E280A ChLNs/ CSs obtained from induced pluripotent stem cells-derived neural progenitor cells” shows that ChLN PSEN 1 E280A and cholinergic CS derived from MenSC, WJ-MSC, and iPSC can reliably reproduce the neuropathology of FAD in vitro, and thus could be cellularly and biochemically equivalent. Furthermore, the study is completed with the dissection of transcriptomic signatures of cholinergic neuronal differentiation using PSEN 1 E280A iPSC, MenSC, and WJ-MSC. The data offered by the authors can provide clues to follow in the search for therapies for AE. Therefore, it would be interesting to publish it with the following modifications.

First: in the anti-plagiarism analysis carried out, a high percentage of coincidences appears (30%), however, most of them belong to material and methods and in some cases to the authors themselves, therefore, the authors must carefully review the section. If there is nothing new in the techniques, perhaps it would be good to reference the article where it was done before and fill it in with the new content.

Second: The title is very long, summarize it and remove the abbreviations.

Third: In the Results section the first figure appears as Figure S1, perhaps before it was from a supplementary material, however, in the text it appears as Figure 1. Make the necessary change.

Fourth: In general, all the figures are too dense, which means that the details of the microphotographs are not appreciated and that the description of the same is exhausting, reaching in some cases the extension up to the Y. Make compositions less dense and with sizes greater than the photomicrographs.

Fifth: The microphotographs of Figures 4, 5, 6, 7, 8, 11 and 12 are too saturated. In the case of the 7, they are also pixelated and some moved. You must modify them, so that they remain as you did. A little contrast can be used, but saturating the photos is not correct.

Sixth: In the materials used, you must add reference, company and country.

Seventh: In the Immunofluorescence analysis section, they say “were fixed with 4% paraformaldehyde for 20 min, followed by Triton X-100 (0.1%) permeabilization..” It seems that the method related to reference 29 has not been followed. I don't understand how they permeate after fixing and without washing. I would appreciate if you could explain it well. If it is a transcription error, modify what is necessary.

Minor editing of English language required

Reviewer 3 Report

An interesting and important research had been carried out by Authors, who offered a new model to study Alzheimer’s disease (AD) in vitro using menstrual blood-derived menstrual stromal cells (MenSCs) and umbilical cord-derived Wharton Jelly’s MSCs (WJ-MSCs). MenSCs and WJ-MSCs might be used to obtain PSEN1 E280A cholinergic-like neurons (ChLNs, 2D) and cerebroid spheroids (3D) which can reproduce familial AD neuropathology more efficiently and faster (11 days) than ChLNs derived from variant iPSCs (35 days). The new in vitro model of PSEN1 E280A familial AD might help to study cholinergic vulnerability and develop curative treatments for Alzheimer’s disease.

The design of the experiment is very logic and well-planned. The obtained results are presented consistently and in details which helps the reader to understand Authors’ idea.

The following comments do not diminish the value of the Article.

Line 25 It would be better to decipher the abbreviature ‘FAD’.

Line 40 Probably it would be better to specify a bit the following keywords: ‘variant’, ‘stromal’?

Graphical abstract is an advantage. Probably it would be better to indicate somewhere that the following numbers: 0, 9, 24, 27, 35, 11 - mean days? But it is very much upon the Authors decision.

Line 271 There is an information about ΔΨm and generation of ROS, probably it would be better to mention it in the title of the paragraph also?

Line 275 It would be better to decipher the following abbreviation: ‘DCF’.

Line 621 It would be better to specify the abbreviations: ‘NM for 0 and 4 days or in mCM’?

The References should be described according the requirements published on the Journal website, would you please check.

Round 2

Reviewer 2 Report

The manuscript has been sufficiently improved to consider publication.